# Estimating Biomass and Carbon Storage by Georgia Forest Types and Species Groups Using the FIA Data Diameters, Basal Areas, Site Indices, and Total Heights

Chris J. Cieszewski [1,*] , Michał Zasada [2] , Roger C. Lowe [1] and Shanbin Liu [3]

[1] D.B. Warnell School of Forest Resources, University of Georgia, Athens, GA 30602, USA; lowe@uga.edu
[2] Faculty of Forestry, Warsaw University of Life Sciences—SGGW, Nowoursynowska 166, 02-787 Warszawam, Poland; Michal.Zasada@wl.sggw.pl
[3] PAREXEL International, 1 Federal Str., Billerica, MA 01821, USA; shangbinl@gmail.com
* Correspondence: biomat@uga.edu

**Abstract:** We present here an example of research into methodology of an estimation of carbon and biomass pools in forests using the USDA Forest Service, Forest Inventory and Analysis (FIA), data of the 1989 and 1998 surveys for Georgia forests, as relevant for comparison with other extremely highly-cited estimates of similar, but different, methodologies. Based on the derived estimates, we produce an example map of the biomass density and pools at a sub-county level resolution, which is based on spatially explicit simulations of the potential cover-type polygons implied by the FIA data with approximate plot locations. Our results include estimates of the biomass pools in the belowground biomass in roots, aboveground woody biomass in trees, and the biomass of foliage. We estimated the biomass densities and pools at a tree level using diameters and heights and previously published models, then propagated these results to the plot level using tree expansion factors, and then transformed these estimates to plot-dependent polygons using plot expansion factors. The plot-dependent polygons were spatially simulated using a simplified assumption of homogeneity of conditions surrounding each plot to the extent of the area defined by this plot's expansion factors. The derived map provides a visual representation of the distribution of forest biomass densities and pools in the state of Georgia with distinctive patterns observed in various areas of urban development, federally owned forests, primary commercial forestland, and other land use areas. Coniferous forests with the highest total biomass density are located mostly in three regions: northern Georgia (Appalachian Highlands), the southern part of Piedmont, and the eastern part of Coastal Plain. Deciduous and mixed forests with the highest biomass density are concentrated mostly in the northern part of the state—especially in the Blue Ridge physiographic province, and in the western part of the East Gulf Coastal Plain. Counties with the highest biomass density were located primarily in the northern part of the state, while counties with the lowest density tended to be located in the Coastal Georgia area.

**Keywords:** aboveground biomass; belowground biomass; biomass distribution; carbon cycle; carbon sequestration; disturbance; hardwood forests; softwood forests; forest inventory

## 1. Introduction

Plant growth produces biomass through photosynthesis absorbing carbon (C) from $CO_2$ in the atmosphere, freeing $O_2$, and thus providing more available oxygen in the air, which is vital for human and animal life and growth. Forests, consisting of tree populations, are particularly significant in biomass production because the growth of wood, tree structural material, takes relatively little organic nutrients while utilizing big quantities of carbon for the tree structural stem growth. For that reason, forests are an important part of global and regional terrestrial carbon cycles, as they can store large amounts of carbon in their biomass and soils. Because of their importance in sequestration of $CO_2$ and carbon

storage, there is a need for accurate and realistic estimation of the amount of carbon they store and sequestrate in a given period of time. Information about amount of carbon is available through estimation of forest biomass, because carbon represents about 50% of dry biomass [1,2].

Accurate and timely inventory of biomass and carbon is especially important in fast-changing areas. This includes changes caused by human activity, such as harvesting and clearing for non-forest uses. Such a situation is known to exist in tropical forests (e.g., [3]) and other developing regions of the world. In the United States, the fastest changes are observed in the southeastern part of the country [4], which can be amplified through intensive plantations management, resulting in increased gains in biomass growth and carbon sequestration [5], as well as more robust sustainability levels of the harvested biomass.

Changes in the management of public lands, particularly in the west, have significantly reduced timber harvest on national forest lands. In response, timber harvests in other regions, particularly in the southeast, are expected to increase markedly in the future.

Georgia is one of the typical southeastern states that in recent decades have been undergoing rapid changes and development. It is the third-fastest growing state in the Union, with over 72 percent of forest cover. Approximately 630 000 private landowners own over two-thirds of Georgia's forests. Georgia has over 9.5 million hectares in commercial forests (approximately 65% of the total of approximately 15 million ha). The value of harvested forest products to the state's economy in 1996 was \$1.22 billion [6]. Because of the importance of forests and their rapid rate of change in Georgia—as well as extensive forest cover, many private owners, forest industry owning large portion of the forestland, and high rate of development—it is an interesting object of study for description of biomass resources and their estimation.

There are relatively few publications concerning the biomass resources of Georgia and other states in the southeastern United States. One of the most detailed analyses of the forest biomass resources in the United States, including individual states, is General Technical Report WO-57 by USDA Forest Service ([7]). All values with this report are expressed as dry biomass, so it is possible to use the presented results to directly estimate the amount of carbon. Another source is the paper by [8], which describes biomass in green pounds and contains detailed statistics of Georgia's forest biomass by broad management classes, diameter distributions, forest types, etc. In 1998, the USDA Forest Service published more information about Georgia's resources [9], but this report contains rather limited information regarding the estimation of biomass and carbon. Biomass analysis is a small part of this 1997 inventory report, which contains only one table with green biomass estimation by ownership class, species group, and tree component. Two more papers on the subject are [2] and [10]. The earlier one describes much broader information about carbon storage and accumulation in the United States forest ecosystems, with separate analyses for the eastern and western parts of the country. On the other hand, refs. [10–12] describe detailed analysis of quantity, and spatial distribution of dry biomass in forests of the eastern USA. These articles, which include consideration of data from Georgia, are based on the use of biomass expansion factors (BEF) applied to the Forest Inventory and Analysis (FIA) survey information from the 6th FIA forest survey of the eastern states that was concluded in 1989.

The aim of this study is to present an alternative methodology to the highly-cited BEF-based methodology proposed by [11]. Contrary to [11], which built on analysis of hardwood data from the tropics and then used diameter-only-based models, we propose the alternative that relies on published models from the USA and the use of both height and diameter data for biomass estimation. In what follows, we describe a sub-county level analysis based on the same 1989 FIA survey data for Georgia, for comparison of the methodology outcomes, as well as on a newer complete forest inventory data for Georgia available from the subsequent 7th FIA survey of 1997, for the illustration of the following changes in the Georgia forest carbon pools. We focus on dry biomass and carbon storage in Georgia's forests for three pools with distinction by various cover types and species groups.

The considered carbon pools are: (i) belowground biomass in roots; (ii) aboveground woody biomass in trees; and (iii) the biomass of foliage.

## 2. Materials and Methods

The most reliable methods of forest biomass and carbon storage calculations are based on using forest inventory data (e.g., [1]). For our analysis, we used the 1989 and 1997 forest inventory data from the largest forest inventory database that is available—USDA Forest Service FIA (Forest Inventory and Analysis)—as the basis for biomass and carbon estimation in Georgia. We used the functions of tree diameters and heights for calculating the total below- and aboveground tree foliage and dry biomass for each measured tree on the FIA plots. When separate equations for foliage biomass were not available, the foliage biomass was calculated by subtracting the total tree biomass estimate without foliage from the total tree biomass estimate with the foliage following the example of [13]. Per-tree values were expanded to per-acre and per-area total tree aboveground biomass using the FIA expansion factors [14,15]. Belowground biomass was estimated consistently with [16], using a model relating the quantities of belowground biomass to the aboveground biomass. The biomass densities per unit area were calculated using appropriate sums of the expansion factors.

### 2.1. Study Site, Biomass, Forest Types, and Species Groups

The study site is the state of Georgia, located in the southeastern United States, which is the part of the country with the fastest changes in forest growth [4]. This research is a part of a larger project on biomass assessment (e.g., [17]) and its production sustainability at different levels of utilization. Below, we describe the underlying assumptions for the biomass estimation in different forest types and species groups.

There are many definitions of biomass used by different authors. The USDA Forest Service uses the notion of "total tree". This is the aboveground portion of tree, without foliage, stump, and roots (e.g., [18]). In practical application of the US forest inventory, the term "total biomass" denotes the total aboveground biomass of a sample tree 2.54 cm (1 inch) diameter at breast height (DBH) or larger. Per tree values must be multiplied by appropriate expansion factors to obtain per area information [14]. This kind of biomass is also called woody biomass (e.g., [8]), and it is an estimate of the amount of economically important goods, which are usually expressed in green pounds. [19] defines "total tree" as bole, top, branches, stump, and roots. Some authors also use the term "complete tree" (e.g., [18]), which is not clearly defined, but it seems to have the same meaning as "total tree". Some publications do not explain the meaning of "biomass", which makes the use of the information in these publications more difficult and subject to erroneous interpretation. In this report, we use the terms "total tree biomass" and "total aboveground biomass without foliage" consistently with the use of this term in the USDA Forest Service publications. These quantities are expressed in dry tons.

Estimation of biomass for roots, foliage, and woody part of tree is desirable, because it reflects differences in properties of tree parts and in their role in the carbon cycle. Thus, for example, tree biomass can be harvested as wood products (e.g., timber, pulp, chips, etc.), while the foliage and root biomass usually remain on site. Further, foliage biomass is decomposed relatively quickly, while stumps and roots are decomposed at a much slower rate. Accordingly, we computed dry biomass of foliage and the stump–root system. Foliage biomass was calculated using a series of regression equations and by subtraction of estimates of biomass without foliage from the estimates of biomass with foliage. Since the root system of the tree comprises approximately 20% of merchantable bole biomass [19], we calculated belowground biomass from aboveground biomass using [16] regression equation. As discussed above, total biomass is defined as the sum of total tree wood, foliage, and root biomass. The amount of carbon was estimated from dry biomass using a factor of 0.5 [1,2] and expressed in Tg ($10^{12}$ g).

The basic unit in the inventory database is a condition on the sample plot, which can be identified as a sampled part of the forest stand. Each condition/plot consists of individual tree measurements, which are subsequently generalized to all trees in the given stand. Any stand is an aggregation of trees, and the stand biomass is defined as the sum of the biomass of the individual trees that comprise the stand [20]. Accordingly, we calculated tree, foliage, and root biomass per acre as a sum of the biomass, calculated with regression equations for each sampled tree, subsequently multiplied by its appropriate tree expansion factor, to obtain the value it represented per unit area. Finally, these values per unit were multiplied by the appropriate plot expansion factors to obtain the values for the summing up into whole state, cover types, and species groups.

For the most important tree species in Georgia (shortleaf and loblolly pine, longleaf and slash pine, red oak, sweetgum, yellow poplar, tupelo-blackgum, and white oaks), we used species-specific biomass equations for the calculation of the dry tree and foliage biomass. The sum of the tree biomass for these species in Georgia accounts for approximately 80% of all woody biomass in the forests [9]. For other species, we used general equations developed for "all" or "other" trees. Equations used for computing the tree dry biomass from tree DBH and tree total height are listed in Table 1.

**Table 1.** Sources of equations for dry biomass calculation.

| Tree Species or Species Group | Source of Equation(s) |
| --- | --- |
| Shortleaf pine | [18] |
| Loblolly pine | [21] |
| Longleaf pine | [13] |
| Slash pine | [22] |
| Other pines—as slash pine | [22] |
| Hardwoods | [23,24] |

*2.2. Data Acquisition*

The USDA Forest Service used to provide the data for all states through the North Central Research Station website, but, currently, the FIA provides the access to data through the FIA DataMart website (https://apps.fs.usda.gov/fia/datamart/, accessed on 21 January 21 2021). This site contains a set of data files in various formats, including the "comma separated values" (CSV) format that can be imported directly into any spreadsheet, database management system, or statistical program. The site contains also the state reports and a detailed manual for the database ([15], based on [14,25] and other manuals). The hierarchically-organized data contained nine files for each inventory: survey, county, plot, subplot, condition, boundary, tree, seedling, and site tree file. This allowed analysis on various levels of resolution (tree, plot, area, county, state, region, and national) by different users (foresters, politics, timber industry).

About 100 features were recorded for each plot, subplot, and condition, including: plot number, ownership, current forest type, stand age and stand-size class, stand origin, site productivity class, site index and site index base age, land use class, basal area per acre, slope, aspect, and, in some cases, elevation, physiographic class, or soil group, treatment opportunity class, percent of unstocked area, stocking, remeasurement period, expansion factors for area, volume, growth, mortality, and removals, location in terms of longitude and latitude, and measurement date.

Over 60 variables were recorded on the tree level. Some of them were collected directly by measurement of trees: tree number, status, species and species group, current and previous DBH, total height, quality class, crown ratio and crown class, and damage and its cause. Other values (e.g., different volumes, volume, removals and mortality expansion factors, or number of trees and number of mortality trees per acre, growth, and biomass) were calculated using formulas [15].

FIA inventories were designed to meet specified sampling errors at the state level at the 67 percent confidence interval. The maximum allowable sampling error for an area of one million acres (404,694 hectares) of timberland is 3 percent. The maximum sampling error for volume and net annual growth on timberland with a billion cubic feet (28.3 million cubic meters) of growing stock is 10% [9]. Using the database for estimation of values on smaller scale (e.g., county level) increases the level of error due to a decrease in the sample size. During estimation of the biomass, a slightly higher error is expected due to an additional source of error from use of regression equations to predict biomass.

*2.3. Biomass Estimation*

Even though a majority of authors agree that using forest inventory data is the most appropriate approach for biomass and carbon sequestration calculations, they use it in different ways. One group of practical methods, represented by Brown [26], Schroeder et al. [11], or Brown et al. [10], use biomass expansion factors (BEF) for converting inventoried wood volume to estimates of above- and belowground biomass. Biomass expansion factors are usually defined as a ratio of aboveground biomass density of all living trees for some predefined merchantable volume. This method is best used for secondary to mature closed forests only. Total aboveground biomass density is calculated as a product of volume per area unit, volume-weighted average wood density, and an appropriate biomass expansion factor. For American forests, BEF values vary from 0.50 to 0.69 [26]. Because forest inventories often report total volume defined in different ways (e.g., merchantable volume only, or volume of trees above a threshold diameter), and these inventories may be the only information available, some authors propose to express volume data in a unified way, or use some common denominator, e.g., volume of trees 10 cm and greater. For example, Brown [26] developed volume expansion factors that related total volume to various merchantable volume estimates. However, use of such an approach can lead to large and unknown errors, especially during extrapolation. Researchers using this approach believe that it is an appropriate method for broad-scale studies, because inventory data are generally collected at large scale from the population of interest and are designed to be statistically valid.

Often, inventories of volume do not characterize total forest biomass well due to the focus on commercial species, measuring trees with diameter bigger than a given threshold, and little or no information on branches, twigs, bark, stumps, foliage, roots, and seedlings and saplings. Another approach to biomass calculation uses biomass equations based on direct tree measurements (diameter at breast height—DBH or DBH and height) that do not require conversion from volume estimates. This approach involves estimating the biomass per tree or average tree of each DBH class, then multiplying the per tree value by the number of trees in the class, and summing tree component estimates for all trees or across all diameter classes. Even though some problems may exist with this method [26], it is a viable approach that provides estimates of the total biomass as well as the various components of it. The FIA database provides for all species detailed measurement of trees with diameters of 2.54 cm (1 inch) and greater. These data can be used to calculate biomass with a smaller number of additional, uncertain assumptions. Existing biomass equations developed for most of the forest species in the southeast made this approach possible for our study. Several papers containing equations for biomass calculations based on diameter at breast height only, and on DBH and tree height, are available in the literature (e.g., [13,18,21–24,27]). There are also papers describing biomass determination based on tree height alone, but this method is valid only for very young trees, e.g., 1–4 years old [28], and they did not apply to our study.

In principle, models available for estimation of biomass from DBH alone can be expected to have large biases depending on site productivity, stand density, and age, because a short tree with a large taper and a tall tree with small taper may have the same diameters but drastically different volumes and biomass contents. The use of both DHB and height is more desirable, when both of these parameters are available, than using DBH

alone, even if the heights are estimated with appropriate functions instead of measured directly. Thus, to compute the biomass and carbon quantities using our approach, we used both tree DBH and total height for each of the measured trees on each plot. While tree heights were not recorded in the 1997 FIA inventory database available online, this database contains the stand site indices, which can be used to estimate height for the dominant trees with appropriate site index equations available for all main eastern forest tree species (e.g., [29]). Heights for all the trees on the plot can be estimated with the models developed for lake states by Hahn [30]. Hahn's model [30] estimates tree height as a function of species-specific site index, stand basal area, and tree DBH. The model has the following algebraic form:

$$H = 4.5 + a \times \left(1 - e^{b \times DBH}\right)^c \times SI^d \times \left(1.00001 - \frac{tdob}{DBH}\right)^f \times BA^g \tag{1}$$

where $H$ is tree height (in feet); $a$, $b$, $c$, $d$, $f$, and $g$ are coefficients (the coefficients for all species are shown in Table A1 in the Appendix A. If a species does not have coefficients, we use coefficients for "other softwoods" or "other hardwoods"); $e$ is the base of the natural logarithm (2.71828); $DBH$ is tree diameter at breast height (in inches); tdob is top diameter over bark (in inches), 0.0 if total height, $SI$ is site index of the stand (in feet, base age 50); and $BA$ is basal area of the stand (in square feet per acre).

Finally, since there may be discrepancies between the height estimates from the local site index models and from model (1), the height estimates for all diameters were multiplied by a ratio of dominant height estimates from the local equations over the dominant height estimates from model (1). This adjustment addresses any potential regional biases that might otherwise underestimate or overestimate heights for any given site index or diameter class, since the shape of site index curves might be different for different regions and species. Use of heights, diameters, and stand densities (through the use of the basal areas), in principle, offers an important improvement over the use of just diameters for tree biomass estimation, because besides the abovementioned principal argument, for most southern pines at the time of doing this study, there were no equations for biomass as a function of DBH alone.

### 2.3.1. Biomass Per Tree

As mentioned above, we found several equations for biomass calculations. Assuming that volume, biomass, and carbon content depend not only on tree diameter, but also on its height and taper, we chose equations based on both DBH and total height. Biomass of softwood species was calculated using the following modified formula, based on [13,18,22], that gives biomass expressed in kilograms based on data provided in imperial units (used by the USDA Forest Service FIA program):

$$Biomass_1 = 0.45359237 \times 10^{a + b \log_{10}\left(DBH^2 \times H\right)} \tag{2}$$

where $Biomass_1$ is a tree biomass expressed in kilograms; a and b are adjusted species or group-specific coefficients (shown in the Appendix A Table A2); $DBH$ is diameter at breast height of the tree (in inches); and $H$ is total height of the tree (in feet).

When necessary, foliage biomass was calculated by subtraction of biomass with and without foliage. Biomass equations for hardwood species are more complicated. Their developers found that the best form of equation depends on tree diameter. In this case, tree biomass of hardwood species trees with DBH below 28 cm (11 inches) was calculated using the following equation [24] with recalculated coefficients:

$$Biomass_2 = a \times \left(DBH^2 \times H\right)^b \tag{3}$$

For trees with diameter at breast height equal to or greater than 28 cm, the following equation was used:

$$Biomass_3 = a \times \left(DBH^2\right)^b \times H^c \tag{4}$$

where: $Biomass_2$ and $Biomass_3$ are tree biomasses expressed in kilograms; $a$, $b$, and c are species or group-specific coefficients (shown in the Appendix A Table A3), and all other symbols are as defined earlier.

Because Equations (3) and (4) cannot be used to predict foliage biomass directly, we obtained these estimates by subtraction, as described earlier.

Most biomass studies, including BEF and allometric equation development, are typically limited to the aboveground tree components, because methods for belowground studies are technically difficult, labor-intensive, and time-consuming. Most existing biomass studies are based on relative biomass allocation between roots and aboveground components—a root/shoot (R/S) ratio. The simplest approach assumes a static relationship for root biomass determination (e.g., [31]), but, in fact, the relationship is most likely highly variable. In fact, we know that root biomass proportions depend on species, soil type, texture and moisture, nutrient availability, etc. Cairns et al. [16] provide a recent review of various root biomass estimation methods. The authors showed, using linear regression analysis, that aboveground biomass, density, age, and plot location (latitude) are the most important predictors of root biomass density. These three factors together explained about 84% of the variation. Comparison of their approach and other methods using R/S ratios for forests in the United States gave about 20% higher estimates. We decided to use this approach since it is relatively simple and useful for the data we had available from the FIA inventory. Root biomass was calculated as function of dry biomass of the tree with foliage [16]:

$$Biomass_{root} = e^{-1.085 \, + \, 0.9256\ln(Biomass)} \tag{5}$$

where all symbols are as previously defined.

### 2.3.2. Total Biomass Calculations

Equations (2)–(5) above allow us to calculate biomass represented by a single tree with a given $DBH$ and total height. These values were expanded first to a single plot area and then to total inventoried area. Total biomass represented by each plot in the entire state inventory was calculated using the formula from Forest Inventory and Analysis database manuals ([14,15]):

$$Biomass_{plot} = Biomass_{tree} \times VOLFAC \times EXPVOL \tag{6}$$

where: $Biomass_{tree}$ is per tree biomass calculated from Equations (2)–(4) given above, $VOLFAC$ is the tree expansion factor (number of trees per area unit that given tree represents in the inventory), and $EXPVOL$ is the plot volume expansion factor (area that given plot represents in the inventory).

### 2.4. Visualization of the Estimated Biomass and Carbon Quantities

At the time of this research, the USDA Forest Service provided only approximate locations of their sample plots to within the nearest 100 s (0.028 degrees), which means precision of this item along the meridian is $\pm1542$ m for latitude and $\pm1094$ m for longitude at latitude 45 degrees [15]. Unfortunately, the inexact locations do not allow for applications of such analysis as kriging, co-kriging, regression, or nearest neighbor analysis. Because of that, we were forced to explore other approaches that are less sensitive to the exactness of the plot locations.

Based on the approximate locations provided by the FIA database, we generated a hypothetical map of Georgia's forests using an algorithm based on what we call the "growing circles" approach. We assumed that, starting from the point with an approximate location, we could build polygons with area equal to the given plot area expansion factor

("number of acres that a given plot represents in current inventory"). The "growing circles" approach is based on a systematic grid of points 200 m apart. Grid points within 100 m of a road, stream, river, or pond greater than 4 hectares, or lake were identified and not processed. The goal of this method was to "grow" (increase size) each circle until its area equaled that of the FIA expansion factor. We computed the weighted distance, using the inverse of the FIA expansion factor, from each grid point to the nearest FIA point and assigned it, and the FIA plot identifier (of the closest FIA point), to each of the grid points. The second iteration involved looping through each FIA point, starting with the point with the smallest expansion factor to select all grid points assigned to each of the FIA points. If the number of selected points was less than the FIA plot area expansion factor divided by 10, then all of those selected grid points were assigned a flag representing the current FIA point, and eliminated from further processing. The number of grid points required was recorded for each inventory plot. If the number of selected points was greater than the number needed, (FIA plot area expansion factor/10) grid points were assigned a flag, starting with the smallest weighted distance. If the latter was the case, we removed that FIA point from further processing. The final iteration utilized only the FIA points that had not been assigned (FIA plot area expansion factor/10) grid points. Starting with the FIA point with the smallest expansion factor, we selected all grid points assigned the current FIA identifier (flag). Iteratively, we selected all grid points without an FIA identifier within 200 m (660 feet) of those selected points. Grid points were assigned a flag until the assigned area was equal to the current FIA point expansion factor.

The resulting grid point dataset included (for each point) the weighted distance to the nearest FIA point, the identifier to the nearest FIA point, and the flag value representing which FIA point it had been assigned. The point dataset was converted to a GRID data type using ArcView's AsGrid request. The GRID dataset was then converted to a polygon using ArcView's AsPolygonFTab request. The final polygon dataset contained 31,503 polygons with each polygon containing a weighted distance, "closest FIA point", and FIA flag attribute.

The result of this approach to spatial population of FIA data is shown in Figure 1. Given that the expansion factors were determined for each plot based on visual inspections of aerial photography, this approach produces a simplified realistic spatial representation of the inventory.

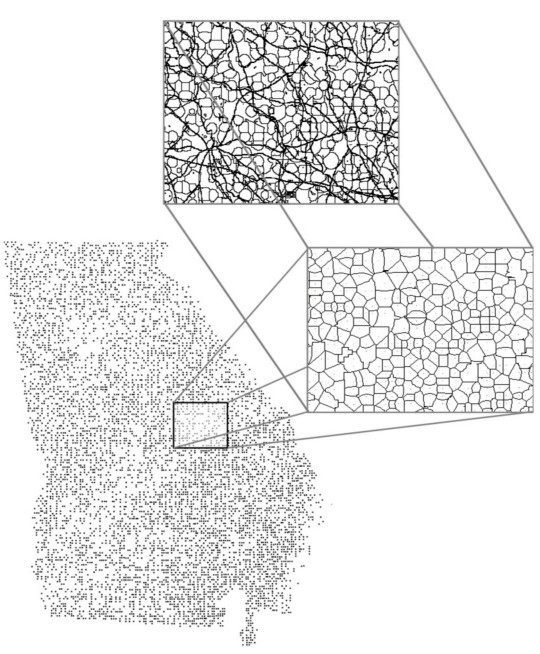

**Figure 1.** Spatial population of Forest Inventory and Analysis (FIA) data based on "growing circle" algorithm.

## 3. Results

Estimated tree dry biomass in Georgia forests in 1989 was 809 million tons, and in 1997 was 844 million tons. Biomass of foliage and roots were 28 and 180 million tons, respectively, for the year 1989, and 30 and 187 million tons, respectively, for 1997. Total biomass of forests (sum of total tree, foliage, and roots biomass) was 1061 million tons. Foliage accounts for less than 3% and roots about 18% of total forest biomass of the state. Total tree dry biomass of 496 million tons was in hardwood species (59% of total biomass). Detailed results with a breakdown by species groups, forest types, and tree parts are summarized in Table 2. Figure 2 provides graphical representation of biomass pools at the county level of resolution.

**Table 2.** Detailed analysis of dry biomass distribution (1989/1997).

|  |  | Tree | | Foliage | | Roots | | Total | |
|---|---|---|---|---|---|---|---|---|---|
|  |  | mil. t | % | mil. t | % | mil. t | % | mil. t | % |
| Species groups | Hardwood | 479 | 59 | 13 | 46 | 105 | 58 | 598 | 59 |
|  |  | 496 | 59 | 14 | 47 | 107 | 58 | 617 | 58 |
|  | Softwood | 330 | 41 | 15 | 54 | 75 | 42 | 419 | 41 |
|  |  | 348 | 41 | 16 | 53 | 80 | 42 | 444 | 42 |
| Forest type | Evergreen | 319 | 40 | 14 | 50 | 74 | 41 | 407 | 40 |
|  |  | 340 | 40 | 15 | 50 | 79 | 42 | 434 | 41 |
|  | Deciduous | 392 | 48 | 11 | 39 | 84 | 47 | 487 | 48 |
|  |  | 394 | 47 | 11 | 37 | 84 | 45 | 489 | 46 |
|  | Mixed | 98 | 12 | 3 | 11 | 22 | 12 | 123 | 12 |
|  |  | 110 | 13 | 4 | 13 | 24 | 13 | 138 | 13 |
| Sum of components' dry biomass (million tons) |  | 809 | 79 | 28 | 3 | 180 | 18 | 1017 | 100 |
|  |  | 844 | 79 | 30 | 3 | 187 | 18 | 1061 | 100 |

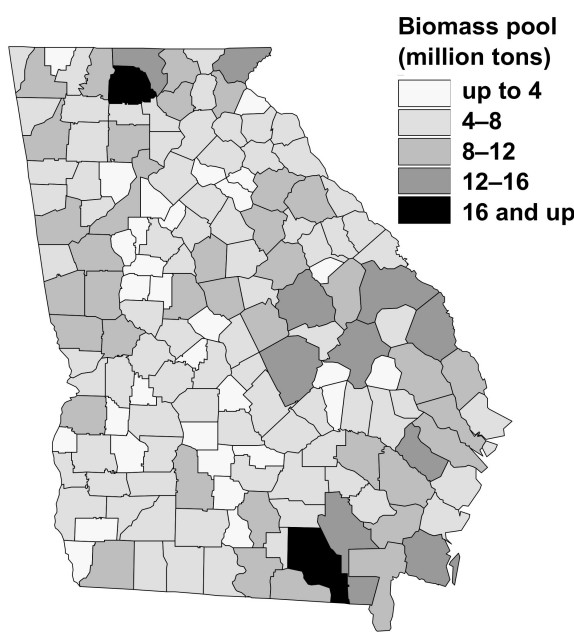

**Figure 2.** Biomass pools (t) on county level of resolution in Georgia.

We estimated the total carbon storage in Georgia forests as 508.5 and 530.5 Tg ($10^{12}$ g) for the years 1989 and 1997, respectively. Similarly, as for the biomass estimation, the detailed results with breakdowns into species groups, forest types, and tree parts are summarized in Table 3. Because carbon is calculated directly from dry biomass, all comments about spatial distribution of the biomass are valid for carbon distribution analysis.

**Table 3.** Carbon pools of Georgia's forests (1989/1997).

| | | Tree | | Foliage | | Roots | | Total | |
|---|---|---|---|---|---|---|---|---|---|
| | | Tg | % | Tg | % | Tg | % | Tg | % |
| Species groups | Hardwood | 239.5 | 59 | 6.5 | 46 | 52.5 | 58 | 299.0 | 59 |
| | | 248.0 | 59 | 7.0 | 47 | 53.5 | 58 | 308.5 | 58 |
| | Softwood | 165.0 | 41 | 7.5 | 54 | 37.5 | 42 | 209.5 | 41 |
| | | 174.0 | 41 | 8.0 | 53 | 40.0 | 42 | 222.0 | 42 |
| Forest type | Evergreen | 159.5 | 40 | 7.0 | 50 | 37.0 | 41 | 203.5 | 40 |
| | | 170.0 | 40 | 7.5 | 50 | 39.5 | 42 | 217.0 | 41 |
| | Deciduous | 196.0 | 48 | 5.5 | 39 | 42.0 | 47 | 243.5 | 48 |
| | | 197.0 | 47 | 5.5 | 37 | 42.0 | 45 | 244.5 | 46 |
| | Mixed | 49.0 | 12 | 1.5 | 11 | 11.0 | 12 | 61.5 | 12 |
| | | 55.0 | 13 | 2.0 | 13 | 12.0 | 13 | 69.0 | 13 |
| Sum of component's carbon content (Tg) | | 404.5 | 79 | 14 | 3 | 90.0 | 18 | 508.5 | 100 |
| | | 422.0 | 79 | 15 | 3 | 93.5 | 18 | 530.5 | 100 |

The estimated average total dry biomass density of Georgia forests in 1997 was 105 tons per hectare. The differences in biomass density were expected between different forest types. Average total dry biomass densities of evergreen, deciduous, and mixed forests were 96, 126, and 91 tons per hectare, respectively. The average biomass density in stands 0–20 years old had an average density of 63 tons per hectare, 21–40 years—111 tons per hectare, 41–60 years old—143 tons per ha, and older than 60 years—178 tons per ha. Total biomass densities at the plot level of resolution mapped, using our "growing circle" approach, are shown in Figure 3. Total biomass densities at the county scale of resolution ranged from 68 to 191 tons/ha as shown in Figure 3.

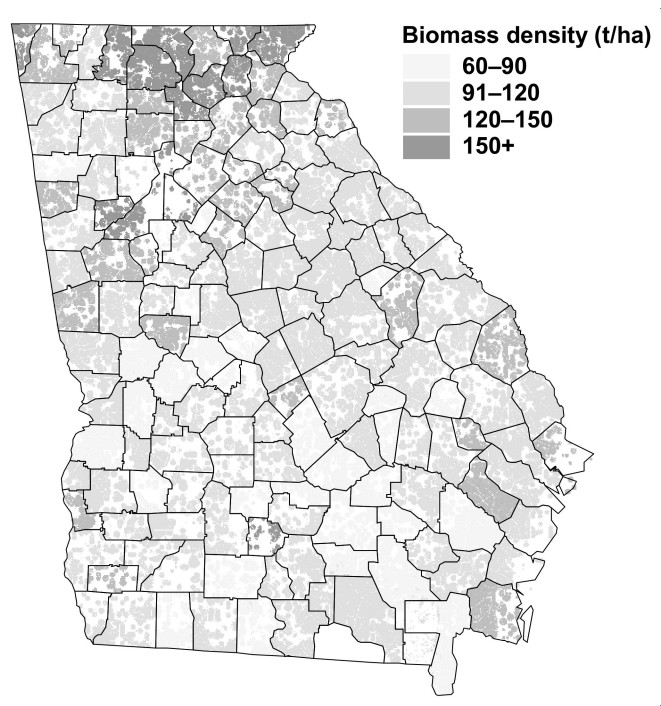

**Figure 3.** Biomass density (t/ha) on county level of resolution in Georgia.

## 4. Discussion and Conclusions

Carbon storage and sequestration estimations are important challenges for ecological, climatic, and economical reasons. The amount of forest carbon stored is relevant to climate change analysis [32], fuel accumulation, and related to it, fire hazard [33], and to forest

sustainability, management, planning, and associated with them, socioeconomic outcomes. Large quantities of biomass imply large quantities of carbon storage, which is, at the same time, a fire fuel accumulation, increasing the risk of fire with corresponding large release of carbon dioxide. Quick growth of biomass implies a dynamic and effective carbon sequestration, removing C from air and freeing oxygen in the atmosphere. For these reasons, the carbon estimation challenge is important, but in no way a new or revelatory problem. Not only have many studies been conducted and many works published over recent decades on the subject, but, in addition, an experienced biometrician can mentally estimate the approximate amounts of carbon in most common forests.

Merely approximating about 10 million ha of Georgia forests with a reasonable maximum yield of about 400 m$^3$/ha (a half of it on average), about 50% of average water content in green biomass and 50% of C content in dry biomass would suggest about 500 Tg of total stored carbon in Georgia forests, which would be within 2%–6% of our detailed computations for the years 1989 and 1997 (Table 3) and within about 6% of the estimate in [7]. Yet, such an estimate, however accurate, is not reliable for estimation of the carbon fluxes in individual pools, and it can be used only for a broad average of the totals. The importance of C storage detailed breakdown into various pools and their fluxes defining C sinks and sources are a strong justification for the need of research into methodologies and improvements of the known methods of C storage estimations. For ecological reasons, we need the ability to consistently compute the carbon sinks and sources based on calculating the changes in biomass over time. This in turn relies on consistency of methodological nuances in underlying assumptions and computations.

A point in case in this study is that, when compared against the Schroeder et al. methodology [11], using heights in biomass calculations is more robust and likely to reflect yield changes over short periods of time than using only diameters for this purpose. It is so because the growth of biomass in natural stands with higher densities is expressed more consistently through the height growth than it is through diameters that can be more likely to be suppressed in high density stands. For that ecological reason, we used, for site quality/productivity classification, the site index, which is a height-based parameter that serves the most reliably as the site quality measure in various forest biometrics challenges of forest management and planning.

Brown et al. [10] found that Georgia had one of the largest biomass and carbon pools in the Southern and Eastern United States. We conducted analysis of biomass and carbon pools in Georgia using the inventory data from the 1989 and 1997 FIA complete inventory surveys and found that the carbon storage of all carbon pools in Georgia forests has further increased from 1989 to 1997.

We found that the hardwoods in Georgia have less biomass of foliage than softwood species (47% of total foliage biomass). Root biomass of hardwoods was calculated using a relationship to total tree biomass, so its proportion is similar to aboveground biomass (58%). The result of the analysis of biomass pool distribution by forest type, which was defined based on plot level data, are similar. Coniferous forests maintain total dry biomass of 434 million tons (41%), deciduous about 489 million tons (46%), and mixed forests 138 million tons (13%). Note that the combined proportion of mixed and deciduous forests is almost the same as the proportion of hardwood species. Overall, our results are similar to those reported by Keays [19].

Calculated total tree biomass was higher than that reported by [7], who calculated total tree dry biomass in 1987 as 745.9 million tons—8% and 12% lower than our estimates of 809 and 844 for 1989 and 1997, respectively (Table 2). The difference may indicate some changes in the increasing biomass storage in Georgia in the years following 1987, but given the big jump between 1987 and 1989, the difference most likely is a result of differences in methodologies. Furthermore, the comparison between these estimates may not be directly applicable since the Forest Service reported the biomass for timberland area only, while we have reported it for all forested areas.

Brown et al. [10] reported 1989 total biomass for the state of Georgia as 1211 million dry tons, which is about 14% higher than our estimate and more than 60% higher than Cost et al.'s estimate [7]. These differences in pool estimates are largely attributed to different methods of biomass calculation applied by the different authors. Brown et al. [10] calculated total tree and foliage biomass using biomass expansion factors (BEF) that convert volume to biomass. In our studies, biomass was directly calculated with regression equations based on measured diameters and estimated from diameter, basal areas, site indices, and total height for each tree. We believe that the methodology presented here, based on the models developed on the local data (USA as opposed to data from tropics), is likely to be more robust than then the other alternatives.

Finally, maps produced in this study allow for easy interpretation of the spatial biomass distribution in Georgia. Figure 4 shows the distribution of total biomass densities by forest types: coniferous, deciduous, and mixed, calculated at the plot scale of resolution. Coniferous forests with the highest total biomass density are located mostly in three regions: northern Georgia (Appalachian Highlands), the southern part of Piedmont, and the eastern part of Coastal Plain. Deciduous and mixed forests with the highest biomass density are concentrated mostly in the northern part of the state (especially in the Blue Ridge physiographic province), and in the western part of East Gulf Coastal Plain. Total dry biomass pools at the county level vary from 1.16 to 19.4 million tons, with an average of about 8 million tons (Figure 2). Counties with the highest biomass and carbon pools are primarily located on the eastern Coastal Plain. This relatively high density is the result of larger forest areas in this region relative to other areas of Georgia. Counties with the highest biomass density are located primarily in the northern part of the state, and counties with the lowest biomass density are located on the coast. Such a distribution of the biomass depends on at least two factors: species composition (forest types) and stand age (Figure 5). The highest densities are in deciduous and mixed forests, which, in addition to having the higher density, are also older than coniferous forests, and the lowest densities are associated with young, coniferous stands, most of which are very young plantations.

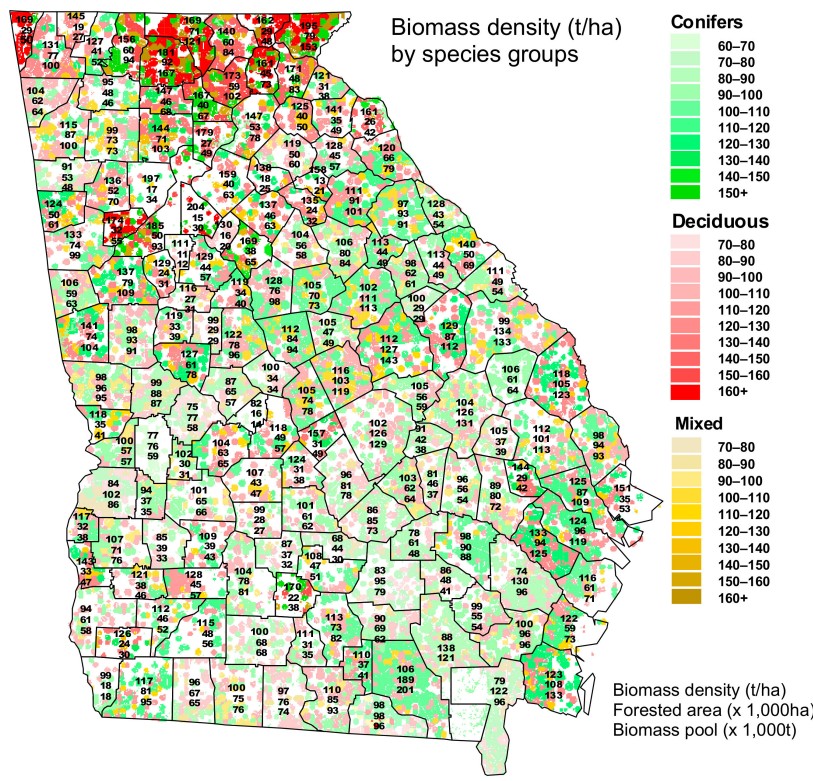

**Figure 4.** Biomass density, county areas and biomass pools on plot scale by forest types (coniferous, deciduous, and mixed) in Georgia.

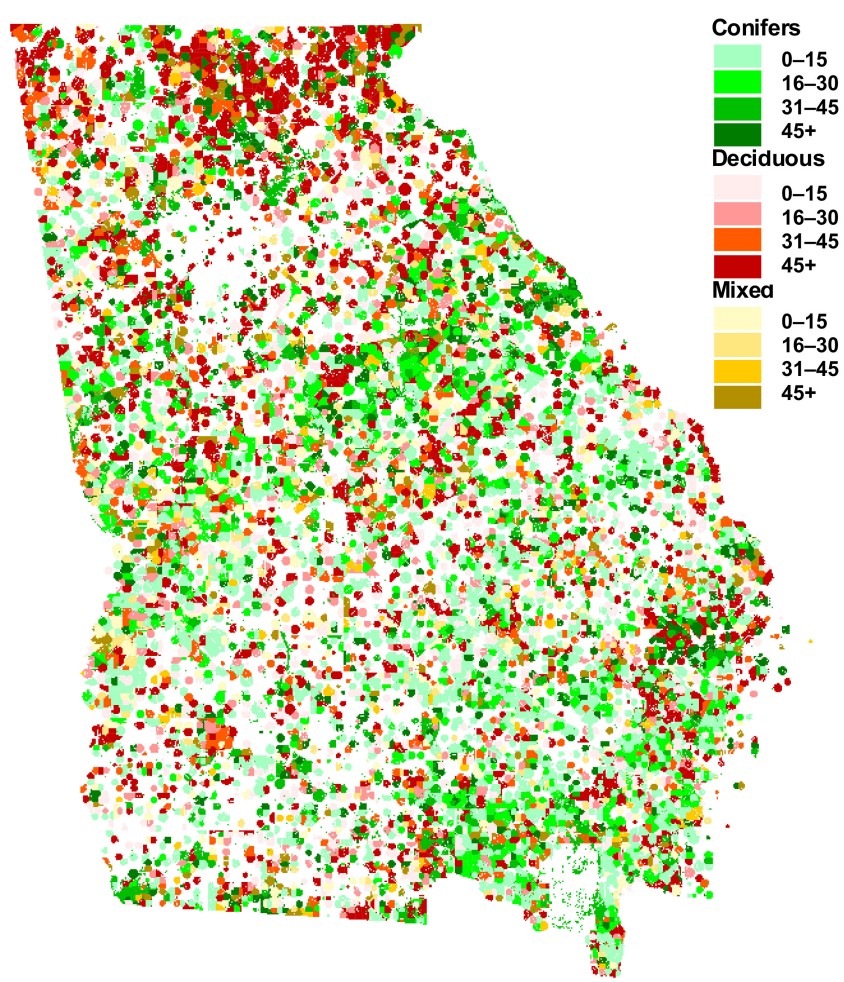

**Figure 5.** Age distribution by forest type in Georgia.

**Author Contributions:** conceptualization: C.J.C.; methodology: C.J.C. and M.Z.; formal analysis: M.Z., R.C.L., and S.L.; writing—original draft preparation: C.J.C. and M.Z.; writing—review and editing: C.J.C., M.Z., R.C.L., and S.L.; visualization: R.C.L.; supervision: C.J.C.; project administration: C.J.C. All authors have read and agreed to the published version of the manuscript.

**Funding:** This research received no external funding.

**Acknowledgments:** This study has been sponsored by the D.B. Warnell School of Forest Resources, University of Georgia. We are grateful to the USDA Forest Service, especially Carol Perry, Joe Glover, Mike Thompson, and Joe McCollum for their help and valuable comments that helped us to use the FIA database and calculate values from the database.

**Conflicts of Interest:** The authors declare no conflict of interest.

## Appendix A

**Table A1.** Coefficients of Equation (1).

| FIA Species Groups | Coefficients | | | | | |
|---|---|---|---|---|---|---|
| | a | b | c | d | f | g |
| Jack pine | 16.934 | −0.12972 | 1 | 0.20854 | 0.77792 | 0.12902 |
| Red pine | 36.851 | −0.08298 | 1 | 0.00001 | 0.63884 | 0.18231 |
| Eastern white pine | 16.281 | −0.08621 | 1 | 0.1622 | 0.86833 | 0.23316 |
| Ponderosa pine | 36.851 | −0.08298 | 1 | 0.00001 | 0.63884 | 0.18231 |
| White spruce | 31.957 | −0.18511 | 1.702 | 0 | 0.68967 | 0.162 |
| Black spruce | 20.038 | −0.18981 | 1.2909 | 0.17836 | 0.57343 | 0.10159 |
| Balsam fir | 14.304 | −0.19894 | 1.4195 | 0.23349 | 0.76878 | 0.12399 |
| Hemlock | 5.3117 | −0.10357 | 1 | 0.68454 | 0.7141 | 0 |
| Eastern cedar, other cedars | 8.2079 | −0.19672 | 1.3112 | 0.33978 | 0.76173 | 0.11666 |
| Other softwoods | 16.934 | −0.12972 | 1 | 0.20854 | 0.77792 | 0.12902 |
| Select white oak, white oak | 9.2078 | −0.22208 | 1 | 0.31723 | 0.8256 | 0.13465 |
| Select red oak | 6.6844 | −0.19049 | 1 | 0.43972 | 0.82962 | 0.10806 |
| Other red oak | 3.8011 | −0.39213 | 2.9053 | 0.55634 | 0.84317 | 0.09593 |
| Select hickory | 6.1034 | −0.17368 | 1 | 0.44725 | 1.0237 | 0.1461 |
| Basswood | 6.3628 | −0.27859 | 1.8677 | 0.49589 | 0.76169 | 0.05841 |
| Beech | 7.1852 | −0.28384 | 1.4417 | 0.38884 | 0.82157 | 0.11411 |
| Hard maple | 5.3416 | −0.23044 | 1.1529 | 0.54194 | 0.8344 | 0.06372 |
| Soft maple | 6.68 | −0.27725 | 1.4287 | 0.40115 | 0.85299 | 0.12403 |
| Elm | 8.458 | −0.27527 | 1.9602 | 0.34894 | 0.89213 | 0.12594 |
| Black ash | 11.291 | −0.2525 | 1.5466 | 0.35711 | 0.7506 | 0.06859 |
| White ash, green ash | 8.1782 | −0.27316 | 1.725 | 0.38694 | 0.75822 | 0.10847 |
| Sycamore | 6.3628 | −0.27859 | 1.8677 | 0.49589 | 0.76169 | 0.05841 |
| Cottonwood, willow | 13.625 | −0.28668 | 1.6124 | 0.30651 | 1.0292 | 0.0746 |
| Balsam poplar, quaking aspen | 6.4301 | −0.23545 | 1.338 | 0.4737 | 0.73385 | 0.08228 |
| Bigtooth aspen | 5.5346 | −0.22637 | 1 | 0.46918 | 0.72456 | 0.11782 |
| River birch, paper birch | 7.2773 | −0.22721 | 1 | 0.41179 | 0.76498 | 0.11046 |
| Black cherry | 5.3416 | −0.23044 | 1.1529 | 0.54194 | 0.8344 | 0.06372 |
| Yl. Pop, Butternut, bl. walnut, | 6.3628 | −0.27859 | 1.8677 | 0.49589 | 0.76169 | 0.05841 |
| Other hardwoods | 6.9572 | −0.26564 | 1 | 0.4866 | 0.76954 | 0.01618 |

**Table A2.** Sources of the original equations and parameters of the biomass Equation (2) for softwood species.

| FIA Species Group | Species | Biomass Type | Reference | a | b |
|---|---|---|---|---|---|
| 110 | Shortleaf pine | Dry without foliage | [18] | −1.55499 | 1.12266 |
| | | Dry including foliage | [18] | −1.52244 | 1.11886 |
| | | Dry foliage | [18] | −2.61282 | 1.03712 |
| | | Green without foliage | [18] | −1.25376 | 1.12517 |
| | | Green including foliage | [18] | −1.20938 | 1.11931 |
| | | Green foliage | [18] | −2.11074 | 1.01076 |
| 131 | Loblolly pine | Dry without foliage | [21] | −1.072 | 0.99421 |
| | | Dry including foliage | [21] | −1.0293 | 0.98788 |
| | | Dry foliage | [21] | −1.87201 | 0.84237 |
| | | Green without foliage | [21] | −0.83678 | 1.01136 |
| | | Green including foliage | [21] | −0.78974 | 1.00404 |
| | | Green foliage | [21] | −1.54968 | 0.83959 |
| 121 | Longleaf pine (DBH $\geq$ 5 inches) | Dry without foliage | [13] | −1.15588 | 1.027 |
| | | Dry including foliage | [13] | −1.06186 | 1.00853 |
| | | Green without foliage | [13] | −0.75522 | 1.00514 |
| | | Green including foliage | [13] | −0.64745 | 0.98442 |

**Table A2.** *Cont.*

| FIA Species Group | Species | Biomass Type | Reference | a | b |
|---|---|---|---|---|---|
| 121 | Longleaf pine (DBH < 5 inches) | Dry without foliage | [13] | −0.71944 | 0.88503 |
| | | Dry including foliage | [13] | −0.65729 | 0.88019 |
| | | Green without foliage | [13] | −0.31359 | 0.85584 |
| | | Green including foliage | [13] | −0.24556 | 0.85263 |
| 111 | Slash pine | Dry without foliage | [22] | −1.20931 | 1.0431 |
| | | Dry including foliage | [22] | −1.16061 | 1.03527 |
| | | Dry foliage | [22] | −1.90538 | 0.85834 |
| | | Green without foliage | [22] | −0.93767 | 1.03929 |
| | | Green including foliage | [22] | −0.88096 | 1.03014 |
| | | Green foliage | [22] | −1.54455 | 0.84989 |
| Other softwoods | as Slash pine | All | [22] | As above | As above |

**Table A2.** *Cont.*

| FIA Species Group | Species | Biomass type | Reference | A | b | c |
|---|---|---|---|---|---|---|
| 812 | Southern red oak (DBH < 11 inches) | Dry without foliage | [24] | 0.06707 | 0.96117 | |
| | | Dry including foliage | [24] | 0.07361 | 0.95348 | |
| | | Green without foliage | [24] | 0.12015 | 0.95457 | |
| | | Green including foliage | [24] | 0.13815 | 0.94228 | |
| 812 | Southern red oak (DBH$^3$ 11 inches) | Dry without foliage | [24] | 0.0277 | 1.14557 | 0.96117 |
| | | Dry including foliage | [24] | 0.0281 | 1.15418 | 0.95348 |
| | | Green without foliage | [24] | 0.04601 | 1.15471 | 0.95457 |
| | | Green including foliage | [24] | 0.04665 | 1.16866 | 0.94228 |
| 611 | Sweetgum (DBH < 11 inches) | Dry without foliage | [24] | 0.049 | 0.94648 | |
| | | Dry including foliage | [24] | 0.05152 | 0.94351 | |
| | | Green without foliage | [24] | 0.10528 | 0.94503 | |
| | | Green including foliage | [24] | 0.1155 | 0.9383 | |
| 611 | Sweetgum (DBH$^3$ 11 inches) | Dry without foliage | [24] | 0.01278 | 1.22662 | 0.94648 |
| | | Dry including foliage | [24] | 0.01409 | 1.2138 | 0.94351 |
| | | Green without foliage | [24] | 0.03175 | 1.19494 | 0.94503 |
| | | Green including foliage | [24] | 0.03517 | 1.18624 | 0.9383 |
| 621 | Yellow poplar (DBH < 11 inches) | Dry without foliage | [24] | 0.0522 | 0.95352 | |
| | | Dry including foliage | [24] | 0.05583 | 0.9482 | |
| | | Green without foliage | [24] | 0.11943 | 0.93782 | |
| | | Green including foliage | [24] | 0.13684 | 0.92608 | |
| 621 | Yellow poplar (DBH$^3$ 11 inches) | Dry without foliage | [24] | 0.03109 | 1.06155 | 0.95352 |
| | | Dry including foliage | [24] | 0.03296 | 1.05809 | 0.9482 |
| | | Green without foliage | [24] | 0.06298 | 1.07125 | 0.93782 |
| | | Green including foliage | [24] | 0.06819 | 1.07131 | 0.92608 |
| 691 | Tupelo | Dry without foliage | [23] | 0.05548 | 0.92453 | |
| | | Dry including foliage | [23] | 0.05696 | 0.92338 | |
| | | Green without foliage | [23] | 0.11048 | 0.9211 | |
| | | Green including foliage | [23] | 0.11539 | 0.91882 | |
| 693 | Blackgum (DBH < 11 inches) | Dry without foliage | [23] | 0.07011 | 0.93057 | |
| | | Dry including foliage | [23] | 0.07335 | 0.92799 | |
| | | Green without foliage | [23] | 0.11712 | 0.94824 | |
| | | Green including foliage | [23] | 0.12331 | 0.94557 | |
| 693 | Blackgum (DBH$^3$ 11 inches) | Dry without foliage | [23] | 0.02912 | 1.11381 | 0.93057 |
| | | Dry including foliage | [23] | 0.0302 | 1.11305 | 0.92799 |
| | | Green without foliage | [23] | 0.0536 | 1.11125 | 0.94824 |
| | | Green including foliage | [23] | 0.05576 | 1.11106 | 0.94557 |

**Table A3.** Sources of the original equations and parameters of the biomass Equations (3) and (4) for hardwood species.

| FIA Species Group | Species | Biomass type | Reference | A | b | c |
|---|---|---|---|---|---|---|
| 802 | White oak (DBH < 11 inches) | Dry without foliage | [24] | 0.05928 | 0.98979 | |
| | | Dry including foliage | [24] | 0.0612 | 0.98969 | |
| | | Green without foliage | [24] | 0.10312 | 0.98415 | |
| | | Green including foliage | [24] | 0.10895 | 0.98258 | |
| 802 | White oak (DBH³ 11 inches) | Dry without foliage | [24] | 0.02926 | 1.13699 | 0.98979 |
| | | Dry including foliage | [24] | 0.03071 | 1.13346 | 0.98969 |
| | | Green without foliage | [24] | 0.0437 | 1.16321 | 0.98415 |
| | | Green including foliage | [24] | 0.05143 | 1.15748 | 0.98258 |
| | Other hardwoods (DBH < 11 inches) | Dry without foliage | [24] | 0.06679 | 0.94275 | |
| | | Dry including foliage | [24] | 0.07153 | 0.938 | |
| | | Green without foliage | [24] | 0.12327 | 0.94274 | |
| | | Green including foliage | [24] | 0.13901 | 0.93307 | |
| | Other hardwoods (DBH³ 11 inches) | Dry without foliage | [24] | 0.02252 | 1.16948 | 0.94275 |
| | | Dry including foliage | [24] | 0.02366 | 1.16867 | 0.938 |
| | | Green without foliage | [24] | 0.04038 | 1.17543 | 0.94274 |
| | | Green including foliage | [24] | 0.04279 | 1.17874 | 0.933137 |

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
