# Peer review of "Estimating Biomass and Carbon Storage by Georgia Forest Types and Species Groups Using the FIA Data Diameters, Basal Areas, Site Indices, and Total Heights"

_forests, doi:10.3390/f12020141_

Round 1

Reviewer 1 Report

Review

General comments
The overall aim of the study is to estimate biomass and carbon stocks for species groups and forest types in Georgia using forest inventory analysis data. Overall I found the topic interesting, though not new. Indeed the study provides new data, on something that is well-known, that biomass and carbon stocks estimation varies between species groups and/or forest types, and that forest inventory analysis data should be preferred for propagating these results to the plot level using tree expansion factors.

The data presented are well-suited to address the aim of the study, and represents an original contribution for under sampled systems. However, I think the sampling is not well written in the paper. For example, the number of plots by species groups and/or forest types is not given and what is the number of trees and species in each plot or forest types?

My major concerns on this manuscript are:

  1. on the writing, which needs to be improved in many aspects, including the overall structure, and more importantly, the study misses ecological background and interpretations. Also, the authors should be more modest on the practical implications.
  2. The statistical analyses are not presented but I suggest testing the differences among species groups and/or forest types in biomass and carbon stocks at the plot level using the analysis of variance and fitting mixed linear models between biomass and structural attributes or environmental attributes as fixed effects and species groups as random effects

Below are some specific and sometimes really minor comments to help clarify the manuscript.

Specific comments

L1-2 please reword the title that is not very informative
L11-13 the very beginning is extremely focused, and a background sentence might be needed.

L18-32 the test for a differential species groups and/or forest types in biomass and carbon stocks is the aim of the study; it has not been demonstrated in this abstract

L37-42 please added the background sentences and references in this first paragraph

L90 please added the aim of this study and the research questions

L91 in the section of Materials and Methods: please reword the overall structure including study site, forest types, species groups and sampling, data collection and biomass estimation, and data analysis.

L366 the discussion and conclusions miss ecological background and interpretations. Please add these informations in the manuscript.

Author Response

Our detailed responses to the Reviewer 1 comments are inserted italicized in within the original comments below:

General comments

The overall aim of the study is to estimate biomass and carbon stocks for species groups and forest types in Georgia using forest inventory analysis data. Overall I found the topic interesting, though not new. Indeed the study provides new data, on something that is well-known, that biomass and carbon stocks estimation varies between species groups and/or forest types, and that forest inventory analysis data should be preferred for propagating these results to the plot level using tree expansion factors.

The data presented are well-suited to address the aim of the study, and represents an original contribution for under sampled systems. However, I think the sampling is not well written in the paper. For example, the number of plots by species groups and/or forest types is not given and what is the number of trees and species in each plot or forest types?

My major concerns on this manuscript are:

  1. on the writing, which needs to be improved in many aspects, including the overall structure, and more importantly, the study misses ecological background and interpretations.

Thank you for the feedback and valuable suggestions. We have revised the manuscript with respect to the comments above, considering the structure of the paper and adding some ecological background without an undue change to the main profile of the paper as relevant to biometrical character of the study and its main aim. We have added some additional ecological context at the beginning of the paper and in the discussion to address the concerns of the reviewer.

  1. Also, the authors should be more modest on the practical implications.

There was no intend in the original manuscript of bragging or overstating the value of this work but to address the above suggestion we have added at the beginning of the Discussion a statement reiterating the views expressed in this review. These views are consistent with our understanding of the statue of this work, which contains a clear disclaimer that the problem of Carbon storage estimation is neither new nor revelatory, but still it is important for ecological, climatic, and economical reasons. This work doesn’t present any ground braking breakthroughs but it arguably offers an improvement of a formerly proposed extremely highly cited methodology with a lot of traction and follow up.

We appreciate the suggestion of doing analysis of variance and a mixed model analysis; however, we didn’t add this components to the revision because it would extent the scope of the study to two papers, while the current manuscript is close to its maximum reasonable length already. There are many additional materials in this study that could be added to the manuscript but due to the length limitations they have been left out already.

The statistical analyses are not presented but I suggest testing the differences among species groups and/or forest types in biomass and carbon stocks at the plot level using the analysis of variance and fitting mixed linear models between biomass and structural attributes or environmental attributes as fixed effects and species groups as random effects

Thank you for the suggestion; however, the aim of the study was to propose improvements in the Carbon storage calculation rather than to test the differences between different subgroups of the calculations. The main premises behind the idea was, and still is, that the local models developed on the data from the USA are better suited for the Carbon storage calculations for the US forests than the tropical data used in the former study. We could test the significance of the differences between the two scenarios but it would not be much of help since even if it showed that the tropical data models worked better for the US forests in this application the conclusion that models based on data from different conditions are better than the local models, would not be generalizable, while a conclusion that local models are better than models developed on foreign data are better, would be an obvious truism.

Below are some specific and sometimes really minor comments to help clarify the manuscript.

Specific comments

L1-2 please reword the title that is not very informative

As per suggestion, we have rewarded the title to: “Estimating Biomass and Carbon Storage by Georgia Forest Types and Species Groups Using the FIA Data diameters, basal areas, site indices and total heights.”

L11-13 the very beginning is extremely focused, and a background sentence might be needed.

As per suggestion, we have rewarded the beginning of the Abstract to make it more comprehensive. As Abstract should contain only the essence of the presented study we kept it still focused on the described study. 

L18-32 the test for a differential species groups and/or forest types in biomass and carbon stocks is the aim of the study; it has not been demonstrated in this abstract

The aim of the study is to propose an improved alternative to an earlier extremely highly cited approach, which still has a lot of traction in the literature.

L37-42 please added the background sentences and references in this first paragraph

As per suggestion, we have added some more background prior to the original beginning of the first section.

L90 please added the aim of this study and the research questions

As per suggestion, we have revised the last paragraph of the first section to explicitly stating the aim of the study. The paragraph now starts with the sentence: “The aim of this study is to present an alternative methodology to the highly cited BEF-based methodology proposed by Schroeder et al. (1997).”

L91 in the section of Materials and Methods: please reword the overall structure including study site, forest types, species groups and sampling, data collection and biomass estimation, and data analysis.

As per request we have rewarded the Materials and Methods section to contain four main sections:

2.1. Study site and biomass in forest types and species groups

2.2. data acquisition

2.3. Biomass estimation

2.4. Visualization of the estimated biomass and carbon quantities

The subsection 2.3 has additional subsections:

2.3.1. Biomass per tree

2.3.2. Total biomass calculations

We didn’t add the additional section of data analysis because in this study the main objective was to compute the quantities rather than do for example the aforementioned significance testing.

L366 the discussion and conclusions miss ecological background and interpretations. Please add these informations in the manuscript.

As per suggestion, we have revised the Discussion section to include ecological background and related to it subjects with added references to other related studies.

Reviewer 2 Report

Dear uthors,

Thanks for the paper on carbon assessment in forests at the local level using available forest inventory data.

The paper combines forest inventory data with available biomass equations for individual tree species and provides tabular and spatial overview of carbon stored in forests of US state Georgia.

Although data from 1989 and 1997 inventories are already outdated, the presented methods may be later applied to newer datasets.

Please find here few suggestions for possible improvements:

L254 - formula 2  - please check appropriate character used for decimal point (the same rule should be applied across the paper, e.g. other formulas and L297).

Employing "spatially explicit polygons" represents a novelty here, however, would be better justified as the forest structures may vary in inventory plot surroundings and thus combination of field inventory data with Earth observation satellite data would provide more reasonable stratification.

Author Response

Our detailed responses to the Reviewer 2 comments are inserted italicized in within the original comments below:

Thanks for the paper on carbon assessment in forests at the local level using available forest inventory data.

Thank you for the help in reviewing the paper and the valuable comments. 

The paper combines forest inventory data with available biomass equations for individual tree species and provides tabular and spatial overview of carbon stored in forests of US state Georgia.

Although data from 1989 and 1997 inventories are already outdated, the presented methods may be later applied to newer datasets.

Thank you. That was the very intend of the paper.

Please find here few suggestions for possible improvements:

L254 - formula 2  - please check appropriate character used for decimal point (the same rule should be applied across the paper, e.g. other formulas and L297).

As per suggestion, we have corrected this typo, which seem to occur only in this formula. The other quantity was correctly delineated by comma as a magnitude separator rather than a decimal.

Employing "spatially explicit polygons" represents a novelty here, however, would be better justified as the forest structures may vary in inventory plot surroundings and thus combination of field inventory data with Earth observation satellite data would provide more reasonable stratification.

Indeed that is true and we have done this kind of visualization in other studies. However, in this study the intend was to illustrate how plot computed numbers could be visualized rather than showing the biomass maps. In many instances the satellite imagery shows “salt and pepper” in plot areas that is more suitable for a kNN imputation rather than the illustration of what a single 1/20 acre plot propagates into.
